Journal of Machine Learning Research 1 (2000) 1-48          Submitted 4/00; Published 10/00

# Automated Quantification Of Blood Microvessels In Hematoxylin And Eosin Whole Slide Images

**anonymous**

**Editor:** Leslie Pack Kaelbling

## Abstract

Tumour cells require resources to survive and proliferate. In order to be provided with a supportive micro-environment rich with resources to sustain optimal growth, tumour cells tend to reside in close proximity to a network of blood vessels. Quantification of blood microvessel density can be a useful measure to investigate the importance of resource limitation in tumours for prognostication and assigning treatment and mode of drug delivery. Currently, immunohistochemistry (IHC) with specific antibodies and the subsequent detection of its binding in the tumour tissue are used to identify microvessels. The automated quantification of blood microvessels in Hematoxylin and Eosin (H&E) stained images is not widely studied because microvessels are very complex and heterogeneous. In addition, their manual identification is tedious, time-consuming and subjective. We investigate whether the vasculature in H&E can be robustly identified in whole slide sections that would ultimately avoid the need for IHC and manual annotations. We propose an artificial intelligence model based on Generative Adversarial Networks (GAN) that, from an input H&E image, can generate a synthetic ERG stained image, highlighting vessel structures. We also trained a spatially constrained Convolutional Neural Network (CNN) to identify single cells on ERG stained whole slide images, and found good concordance between detected cells in synthetic and real ERG. This pipeline was evaluated on 2002 image patches of size 2000×2000 pixels, sampled from 9 whole slide images. We achieved the mean $R^2$ of 0.70±0.14 in our testing set. This pipeline can pave the way to study proximity of tumours cells to blood vessels. This approach has the potential to reduce the use of IHC and tissues and enable large quantitative studies.

**Keywords:**   Blood microvessel density, Generative Adversarial Network (GAN), Image synthesis

## 1. Introduction

The tumour micro-environment (TME) is the tissue structure surrounding a tumour that includes blood vessels, immune cells, fibroblasts, signaling molecules and the extracellular matrix. Tumour cells are spatially and temporally heterogeneous and interact with each other and with the micro-environment. The combination of genetic diversity of neoplastic cells and the tumour micro-environment shape tumour development and progression (Nawaz et al., 2019). TME can be quantified through the concepts of ecology, characterised by hazards and resources available to the neoplastic cells. Example sources of hazards for neoplastic cells include immune cells, toxins, and anti-cancer therapies. While resources, include oxygen, glucose, micro-nutrients, space, survival and growth signals (Reynolds et al., 2020). Different combinations of hazards and resources can produce different fitness landscapes, and thus will have a critical impact on the future evolution and behaviour of cancer cell

populations and outcome for patients. For example, high abundance of $CD^+20$ B-cells and $CD^+8$ T-cell within the tumour promotes better prognosis in ovarian cancer (Nielsen et al., 2012). Currently little is know about the interactions between cell metabolism and the availability of key resources. There are various methods to measure factors that affect resources that may be prognostically relevant. For example, the proportion of a tumour that is hypoxic, blood vessel density, and co-localisation of tumour cells with fibroblast. Of these, we are interested in the blood (micro)-blood vessel density (BMVD).

From clinical point of view, a link between angiogenesis and tumour invasion or metastasis, may suggest that preventing new vessels from forming could be a way to inhibit further tumour growth. The presence of necrosis and hypoxia in many tumours, attests to the importance of resource limitation in preventing tumour growth (Lugano et al., 2020). Evidence indicates that tumour angiogenesis identified by elevated BMVD is associated with poor disease free survival in stage II colon cancer, endometrial cancer and testicular germ cell tumours (Sjoerd et al., 2019; Wang et al., 2018a; Gilbert et al., 2016).

Despite some promising leads, the field will only develop if these biomarkers can be validated in large clinical trials across different tumour types. BMVD is typically identified using immunohistochemistry (IHC) (Haber et al., 2015) and scored manually or by digital morphometric analysis of IHC whole tumour sections. Erythroblast Transformation specific related gene (ERG) is a highly specific endothelial IHC cell marker (Haber et al., 2015). Performing a large scale study with IHC and manual scoring is likely to be very challenging. This is because the production of IHC sections for a large scale trial is likely to be expensive and time consuming. In addition, the involvement of multiple pathologists is likely to be necessary to resolve issues of inter-observer variability, which is difficult to achieve in a field with global staffing shortages. Therefore, it is beneficial to develop automated approaches that can quantify BMVD in histopathology images for clinical association analysis in large scales.

To train supervised models capable of automatic quantification of BMVD in Hematoxylin & Eosin (H&E) images, there is a crucial need for large amount of example images. Manual annotation of example images is very laborious, and the variability of size and shape of the vessels may make producing a good training dataset difficult. We propose a Generative Adversarial Network (GAN) mapping H&E to IHC, allowing us to synthesise or predict ERG stain. Subsequently, we investigate whether the use of a synthesised IHC can help identify the vasculature and BMVD quantification to reduce the need for IHC and manual annotations and only use H&E for large scale studies.

## 2. Related Work

A review of the literature indicates that there are, to our knowledge, only two works on automatically measuring the BMVD in H&E images, both based on the segmentation of microvessels. Yi et al. (2018) developed an automated microvessel segmentation algorithms for H&E stained images using fully convolutional neural networks. The feasibility of the proposed algorithm was demonstrated through experimental training using 20 H&E stained Whole Slide Images (WSIs) from patients with lung adenocarcinoma. Their segmentation model achieved the mean pixel accuracy of 0.83 on 35 image tiles (extracted from 5 WSIs). Then, they applied this model to identify the micro vessels in the pathology images of 88

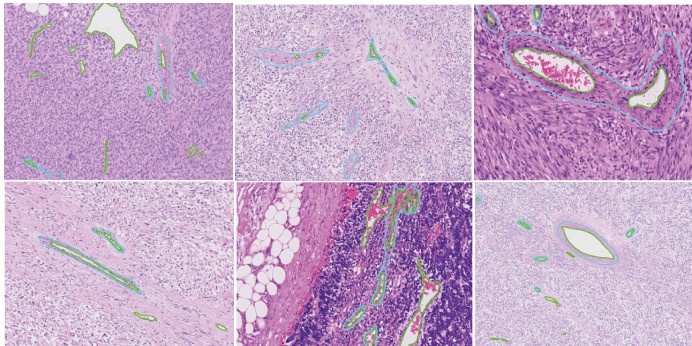

Figure 1: Variation of vessel appearance. Green boundaries represent lumen and blue boundaries represent wall of vessels. Some vessels only have lumen, some only walls and some have both of them. endothelial cells form the interface between the lumen and the vessel wall and wille exist where there is a microvessel.

lung adenocarcinoma patients. They showed that the identified microvessel features (the abundance of BMVD and being surrounded by tumour cells) were significantly associated with the patient clinical outcomes.

The correlation between lymphovascular invasion, tumour angiogenesis and patient survival was also studied by Fraz et al. (2018). They proposed a framework for microvessel segmentation of H&E stained histology images using a segmentation model with depth-wise separable convolutions, followed by spatial pyramid pooling at multiple scales. In addition, an uncertainty prediction mechanism was designed to account for the uncertainty of the model prediction. This pipeline was evaluated on 13 WSIs of oral squamous cell carcinoma tissues. Quantitative performance measures of microvessel segmentation using the proposed model was 0.939 in terms of dice score and 96.94% segmentation accuracy, on images of size 514×514 pixels.

These models were generally confined to small visual fields for input and thus were only capable of segmentation on regions of this size. Therefore, if a microvessel region is larger than the defined visual field, it may not be segmented correctly. Blood vessel phenotype is markedly heterogeneous within a single tumour with regards to their structure, morphology, size and distribution in cross sectional images. This appearance variety leads to a significant class imbalance problem in the training dataset. Some example images illustrating the degree of variation in the appearance of vessels are shown in Fig.1. Each type of vessel has a lumen, wall and endotheliums that line the interior surface of vessels and form the interface between the lumen and the rest of the vessel wall. In the mentioned studies, they focused on segmenting just the lumen of the vessel, however the presence of lumens only does not characterise all types of microvessels in the tissue section. Relying only on H&E samples, limits the study to the texture of the vessels containing visible red blood cells or visible lumens. The other point to mention is that in these studies, the evaluations were performed on the small visual fields rather than whole slide images.

A better way to look for microvessels across the whole slide sections could be through synthesising IHC from H&E.

## 3. Material

20 cases of rhabdomyosarcoma (RMS) were selected, which represent the subtypes of RMS that include both Fusion Positive (PAX3/7-FOXO1) and Fusion Negative cases. Sections were cut and stained with H&E, using standard procedures. The H&E sections were scanned using Hamamatsu NanoZoomer S210. All slides were scanned at $40\times$ magnification with pixel resolution of 0.2204 microns $\times$ 0.2204 microns. Following the scanning, the H&E was heated with acid alcohol to remove the H&E staining and allow for the next procedure. The sections were re-stained with the ERG antibody using standard IHC. The slides were re-scanned. Re-staining, instead of serial sections, results in H&E and IHC WSI pairs that contain the exact same tissue.

Although the image pairs were acquired from the same glass slides, minor alignment errors and tissue deformations were still present due to the re-staining procedure. To address the alignment issue and generate precisely aligned tiles, each H&E-ERG whole slide pair was initially co-registered using a manual point-based registration approach. A linear mapping method based on Affine transformation was calculated for each pair to compensate likely shearing, scaling, rotation and translation, while preserving collinearity and ratios of distances in H&E and ERG-stained whole slide images. Following registration, the images are divided into tiles of 2000×2000 pixels which, due to the previous alignment step, are highly concordant between H&E and ERG and provided us with a clear ground truth for endothelial cells.

## 4. Methodology

We introduce a pipeline that is adapted to high resolution images (2000×2000 pixels at $20\times$ magnification) so that a large and informative field of view can be investigated in the training phase. This visual field can contain varied vessel appearances. We use a synthesis model to generate and predict ERG stains from H&E images resulting in positive endothelial cells rather than segmenting vessels.

For this work, we created two models. First, a generative adversarial model named "ERG synthesis model", which was adapted for larger visual fields was trained to generate synthesised IHC with ERG stain. Subsequently, in order to identify and quantify the location and abundance of endothelial cells (marked positive in the synthesised image), we trained another model to localise the centre of nuclei for all individual cells and then classify each cell type to identify ERG positive cells that correspond to endothelial cells and vasculature.

### 4.1 ERG Synthesis Model

This model (inspired by Wang et al. (2018b)) is a conditional generative adversarial model that has been adapted to learn a coarse to fine mapping of H&E to ERG-stained images in a multi-scale architecture. It is composed of two generative sub-networks and a multi-scale discriminator. The internal generative network operates at a lower magnification (1x) while the outer generative network outputs an image with a higher magnification (10x). This process is similar to zooming in or out to have a global or detailed view of the image by pathologists. This multi-resolution pipeline is used to effectively aggregate global

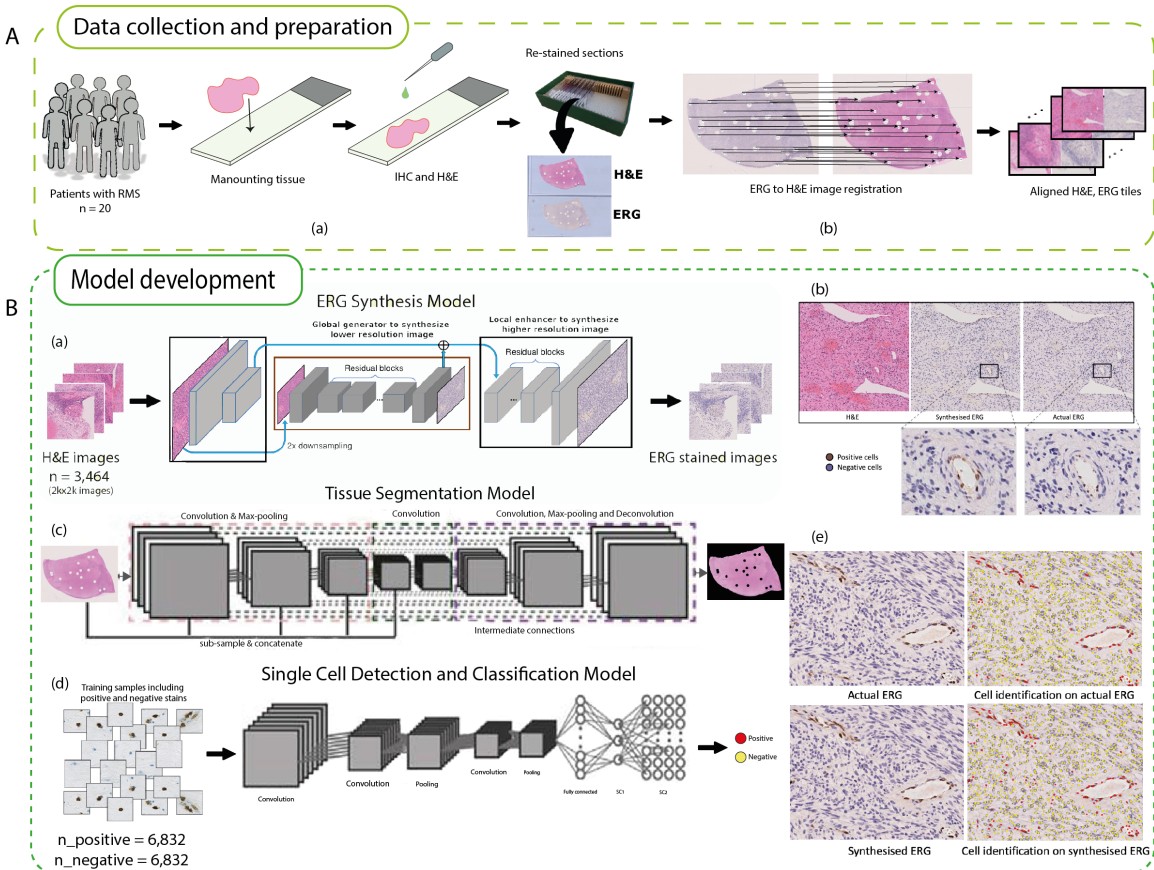

Figure 2: The proposed pipeline: (Aa) 20 cases of Rhabdomyosarcoma were re-stained with H&E and IHC with ERG stain, (Ab) registration was performed on down-scaled whole slide images and then visually checked to prepare precise pairs of H&E and IHC, (Ba) ERG synthesis model that was trained using the prepared paired images, (Bb) an example 2000×2000 H&E image tile along with the synthesised ERG stain and actual ERG stain to compare visually, (Bc) pre-trained tissue segmentation model on H&E images, (Bd) single-cell identification model to localise and classify the centre of nuclei within the whole tissue and (e) output example of the single cell identification model on the synthesised ERG and actual ERG.

(particularly for larger vessels) and local information (particularly for small microvessels) from the images and render H&E stain to ERG stain, preserving the tissue architecture. The overall objective of the generators is to synthesise realistic ERG-stained images from H&E stained images, while the discriminator (used in the training phase) aims to distinguish real ERG-stained images from the synthesised ERG-stained images.

In the ERG Synthesis Model, the generator is composed of two sub-networks: global generator (G1) and local enhancer (G2). The global generator operates at a resolution of 512×512 pixels, and the local enhancer outputs an image with 1024×1024 pixel resolution.

So, the input of the model is an H&E image that will firstly be given to G1. This global generator consists of 3 components: a convolutional front-end, 8 sets of residual blocks and a transposed convolutional back-end. Thus, the input H&E image is passed through the 3 components sequentially to output an ERG-stained image. Likewise, the G2 network consists of 3 components: a convolutional front-end, a set of residual blocks, and a transposed convolutional back-end. The resolution of the input image to G2 is 1024×1024. Different from G1, the input to the residual block is the element-wise sum of the output feature map of convolutional front-end from G2, and the last feature map of the back-end of G1 in order to effectively integrate the global information from G1 to G2. The output of G2 will be the expected synthesised ERG stain. In the training phase, this output was sent to the discriminator to be compared with the real ERG stain and guide the generator to generate finer details. The discriminator is composed of three identical networks where each one operates at different image scales, creating an image pyramid of three scales. The network that operates at the lowest scale has the largest receptive field in order to guide the generator to generate globally consistent images.

The training dataset is prepared as a set of pairs of corresponding images $(HE_i, ERG_i)$, where $HE_i$ is an H&E image and $ERG_i$ is the corresponding ERG-stained image. 20 scanned slide pairs were split into the training (11 cases) and testing (9 cases) sets. After whole slide registration and tissue segmentation steps, we acquired 3,464 pairs from 11 whole slide images to train the model for 200 epochs. The testing set was used as a hold-out set for the evaluation step. Notably, this model uses larger visual image views compared to the previous segmentation-based methods, where the training occurs to result in disconnected segmented regions.

## 4.2 Single Cell Detection and Classification Model

This model consists of three main parts:

- A pre-trained tissue segmentation model on H&E images (AbdulJabbar et al., 2020) (inspired by Micro-Net27 Raza et al. (2019)) is used to exclude the regions of the slide that are unlikely to contain cells, such as background, noise and artefact, reducing processing time on later steps. Each whole slide image was reduced to 1.25× resolution and segmented for tissue regions using multi-resolution input/output image features. Each image was analysed at multiple resolutions by concatenating context information from intermediate deep layers and using bypass connections to maintain features related to the tissue boundary.

- A single-cell detection model is used to localise the centre of nuclei for all individual cells within the whole tissue. A Spatially Constrained Convolutional Neural Network (SCCNN) (Sirinukunwattana et al., 2016) was trained to predict the probability of a pixel being the centre of a nucleus. Using this network, we trained the single cell detection model for our IHC samples (on a combination of real and synthesised ERG-stained cells). The main objective of this step was to detect all nuclei in a whole slide image by locating nuclei centre positions, regardless of their class labels.

- A cell classification model is used to classify each cell type that was detected in the previous step. In this part, we used the Softmax SCNN for the nucleus classification.

This classifier uses neighbouring ensemble prediction combined with the standard softmax for the classification. The main objective of this part was to classify previously detected nuclei into positive (resembling nuclear DAB staining) or negative.

For both detection and classification networks, the SCCNN network consisted of input layer (with 31×31 pixels), convolutional layers, non-overlapping spatial max-pooling and fully-connected layers along with parameter estimation and spatially constrained layers. These SCCNN models were trained in a supervised manner based on single-cell annotations.

For the cell detection model, 509,839 single cell annotations were collected from 20 whole section images (10 WSIs from the real ERG cohort and 10 WSIs from the synthesised ERG cohort). Training data collection for the classification model followed a similar process to that of the detection model. 21,337 annotations were collected from positive and negative cells from 20 WSIs. We used the proportion of 70%, 10% and 20% for creating training, validation and testing sets, respectively based on the WSIs.

## 5. Results

ERG stain synthesis from H&E stain images is a one-to-one mapping problem and an ideal model should generate, from the H&E image, an image that appears positively stained wherever ERG positive cells are present. To validate the performance of the proposed pipeline, we assessed the performance of the (i) cell identification model and (ii) the whole pipeline. The performance of the cell detection and classification model was evaluated on 9 H&E WSIs with matched real ERG and synthesised ERG. We used the same testing cases in the assessment of ERG synthesis and cell identification models. No further post-processing was performed for the synthesised outcome. The SCCNN model's performance on 2,133 cells achieved the F1-Score of 0.9906±0.0087, Sensitivity of 0.9863±0.0175, Specificity of 0.9910±0.0207 and Accuracy of 0.9834±0.0147. The quantitative evaluation of the whole pipeline for the testing cohort is shown in Fig. 3. The performance of the whole pipeline was evaluated using two metrics:

- Morisita Horn Index (MHI) (Maley et al., 2015) to quantify abundance and co-localisation of synthesised ERG and actual ERG stained cells. Each 2000×2000 image tile was divided into 200×200 spatial regions, for each aligned real-synthetic image pair, the MHI between them was calculated. The average MHI was reported for each tile across the whole image slides from the testing set. Our results in Fig 3.A. shows that the proposed pipeline was able to learn and identify endothelial cells with average MHI of >80%, considering the variability that exists in their morphology and distribution. The higher the value of MHI, the higher the overlapping between synthesised and real ERG stained cells.

- $R^2$ to show the correlation between the numbers of cells detected in the real ERG image and the equivalent region generated from H&E. We computed the abundance of ERG positive cells within 2000×2000 spatial regions for synthesised and real ERG and plotted the count of each tile for each case in sub-plots. Given the minimum $R^2$ of 0.46 and the maximum of 0.88, the achieved $R^2$ values were high.

The existing variability in our quantitative results is due to the existing stain variability in the H&E images. When an H&E section suffers from low stain quality, the generated

ERG synthesis can go pale, resulting in false negative and false positive staining. Likewise, over-staining leads to uncertain and sometimes messy prediction of ERG stain. These issues could be solved by training on a larger and more variable training dataset.

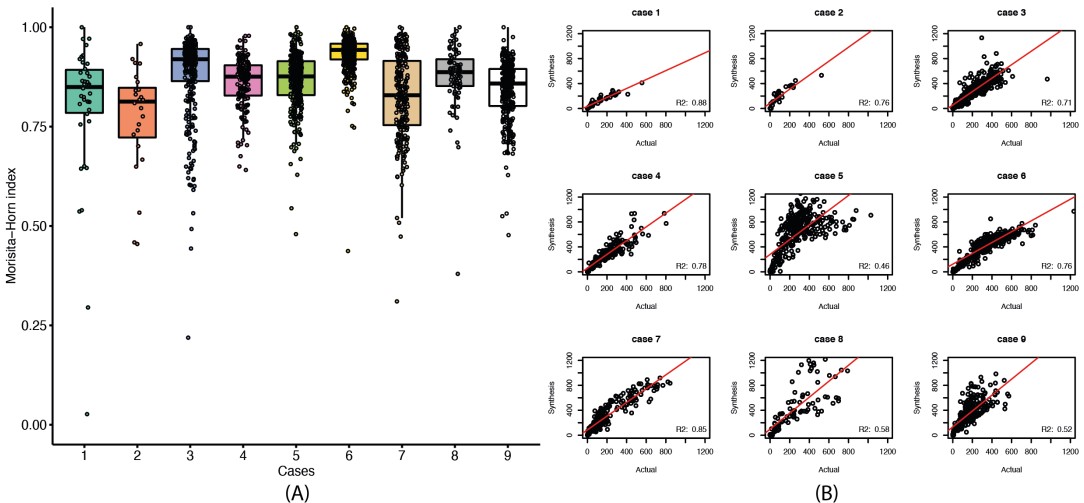

Figure 3: Quantitative results for 9 cases in the testing cohort. (A) MHI and (B) Abundance of detected positive ERG stained cells along with the calculated $R^2$. Each point represents the calculated value within 2000×2000 image tiles. Therefore, the number of points for each case represents the number of tiles that contained tissue after tissue segmentation.

## 6. Conclusion and Future Work

Tumour cells require vasculature to provide oxygen, nutrients, and a means of waste disposal in order to grow. They select the physiological processes of angiogenesis to recruit endothelial cells and a blood supply. H&E and IHC are techniques used throughout pathology to identify BMVD. Manual delineation of microvessels for bio-marker analysis in IHC or H&E images is tedious, time consuming, not reproducible and subjective. We proposed and tested a pipeline that can predict individual cell ERG-stain from H&E stained images. The utility of this pipeline could be in clinical trials to investigate: (i) the effect of resource limitation in tumours by co-localisation of hypoxia markers with vessels (Bernauer et al., 2021) or co-localisation of immune cells with vessels; (ii) morphological changes of vessels and tumour angiogenesis influenced by treatments like radiotherapy over time and anti-angiogenetics; (iii) response to anticancer drugs and also (iv) their association with prognostic markers and disease free survival. Moving forward, we will use other cancer cohorts to investigate the applicability and generalisability of this model and potentials for future improvements. We will use cases from samples of soft tissue sarcoma, glioblastoma, breast and ovarian cancer to extend the applicability of this pipeline.

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
