# OpenReview forum: "Automated Quantication Of Blood Microvessels In Hematoxylin And Eosin Whole Slide Images"
_MICCAI.org/2021/Workshop/COMPAY — COMPAY 2021_

### Official Review · Reviewer_yz5V · 2021-08-03
**This is an interesting paper in which the authors address the question of blood microvessel detection**

**Rating:** 8
**Confidence:** 2

**Review:**

This is an interesting paper in which the authors address the question of blood microvessel detection. To this end, the authors propose a GAN that transform an H&E image into an ERP (IHC) stained image. They report reasonable result.

The methods and data are described and all appear sufficiently clear. (Here and there there are typos or unlucky grammatical constructions but nothing a spellcheck cannot resolve)

My main question pertains to why the authors aim to first transform the H&E image into an ERP stain. If this step works, and the experimental H&E and ERP images are correctly registered, a blood vessel detection model an be directly trained from H&E images.  However, the work itself remains valid and interesting in its own right.

---

### Official Review · Reviewer_jVsx · 2021-08-25
**ERG image synthesis from H&E for blood microvessel quantification**

**Rating:** 7
**Confidence:** 4

**Review:**

This paper proposes a generation of synthetic ERG stained images to highlight vessel structures. A combination of models is then trained to detect and classify cells into different types.
It is well written, easy to follow and well motivated. The results seem promising (the R^2 less than the SCCNN model’s performance unfortunately).

The title seems misleading or some evaluation is missing to actually quantify the blood microvessel density and not only detect and classify cell types.

The qualitative evaluations are too small (in Fig. 2).

Minor comments:
- SCCNN end of page 6
- Fix caption in Fig. 1

---

### Official Review · Reviewer_4dWx · 2021-08-25
**An innovative method for evaluating blood microvessels in H&E images**

**Rating:** 10
**Confidence:** 4

**Review:**

The authors present an innovatie study for recognising and quantifying blood microvessels from ERG images generated from H&E slides.
Their method is novel, based on a small but high-quality dataset, and the paper is well written.
My only methodological concern is the relatively small number of cases included. This applies particularly to the variability of H&E and IHC stains, and the authors should add external validation on a larger dataset to their list of future work presented in Sec. 6.

Further minor points that might help improve the presentation further.
- Define "ERG" before first using it in the abstract
-Fig 3 is quite crowded, and the labels are difficult to read, please improve

---

### Decision · Program_Chairs · 2021-08-25

Accept